# Mutant p53 as an Antigen in Cancer Immunotherapy

**DOI:** 10.3390/ijms21114087

**Published:** 2020-06-08

**Authors:** Navid Sobhani, Alberto D’Angelo, Xu Wang, Ken H. Young, Daniele Generali, Yong Li

**Affiliations:** 1Section of Epidemiology and Population Science, Department of Medicine, Baylor College of Medicine, Houston, TX 77030, USA; Xu.Wang@bcm.edu; 2Department of Biology and Biochemistry, University of Bath, Bath BA2 7AY, UK; ada43@bath.ac.uk; 3Department of Pathology, Duke University School of Medicine, Durham, NC 27708, USA; ken.young@duke.edu; 4Department of Medical, Surgical and Health Sciences, University of Trieste, Cattinara Hospital, Strada Di Fiume 447, 34149 Trieste, Italy; dgenerali@units.it

**Keywords:** p53, serum antibodies, tumor suppressor, immunoncology, cancer

## Abstract

The p53 tumor suppressor plays a pivotal role in cancer and infectious disease. Many oncology treatments are now calling on immunotherapy approaches, and scores of studies have investigated the role of p53 antibodies in cancer diagnosis and therapy. This review summarizes the current knowledge from the preliminary evidence that suggests a potential role of p53 as an antigen in the adaptive immune response and as a key monitor of the innate immune system, thereby speculating on the idea that mutant p53 antigens serve as a druggable targets in immunotherapy. Except in a few cases, the vast majority of published work on p53 antibodies in cancer patients use wild-type p53 as the antigen to detect these antibodies and it is unclear whether they can recognize p53 mutants carried by cancer patients at all. We envision that an antibody targeting a specific mutant p53 will be effective therapeutically against a cancer carrying the exact same mutant p53. To corroborate such a possibility, a recent study showed that a T cell receptor-like (TCLR) antibody, initially made for a wild-type antigen, was capable of discriminating between mutant p53 and wild-type p53, specifically killing more cancer cells expressing mutant p53 than wild-type p53 in vitro and inhibiting the tumour growth of mice injected with mutant p53 cancer cells than mice with wild-type p53 cancer cells. Thus, novel antibodies targeting mutant p53, but not the wild-type isoform, should be pursued in preclinical and clinical studies.

## 1. Introduction

### 1.1. Discovery of p53

Approximately 70 years ago, several DNA viruses such as adenovirus, human Epstein–Barr virus, polyoma and SV40 were found to be able to cause tumors in humans and rodents [1]. In all of these cases, the viral proteins which stem from the viral genome—either integrated into a chromosome or as plasmid—were observed to be involved in tumor promotion, formation and maintenance [2,3,4]. These viral proteins—subsequently named tumor antigens (TA)—were recognized by the immune system and the different antibodies were found to target them specifically [5]. Given this scenario, in 1979, four different groups in England, the United States, and France nearly simultaneously discovered the p53 protein in normal and cancerous cells, testing the serum from animals with spontaneously derived or SV40 virus-induced tumors [6,7,8,9]. Among these four studies, the same 53kD protein (called p53) was detected—and confirmed with peptide maps—in SV40-transformed cells and malignant cells that are not transformed by a virus, whereas decreased p53 levels were observed in uninfected normal cells. Malignant cells that were not transformed by any virus also had increased levels of p53, suggesting that the SV40 tumor antigen, a well-known factor for the tumor initiation and progression, binds to p53 and raises its concentration above its normal levels in control healthy cells [10]. These preliminary results paved the way for a vast number of studies on the role of cellular protein p53 in the cancer biology field, leading to the discovery that p53 mutations are the most common genetic alteration in human cancers.

### 1.2. Tumor Suppression Role

p53 has a unique and unequivocal tumor suppression role, which has been confirmed by the cancer susceptibility of individuals affected by Li–Fraumeni syndrome, the p53 inactivation in most sporadic human cancers, and the spontaneous tumorigenesis in mice with the p53 gene knocked out [11]. During tumor development, inherited and/or sporadic TP53 genetic missense mutations are normally followed by a loss of heterozygosity (LOH), turning into an entire p53 deficiency. It seems there is a selective advantage towards the loss of the remaining allele of the wild-type p53 (p53-wt) gene [12,13,14]. The loss of p53 gives way to the initiation and progression of malignancies, which are generally characterised by more malignant features such as intensified invasiveness and metastatic capability, genetic instability and poor cellular differentiation [15,16,17,18]. In all likelihood, these outcomes are given not only by the loss-of-function (LOF) of wild-type p53 (p53-wt) but also by the tumorigenic gain-of-function (GOF) features of some p53 mutants (p53-mut) described later.

p53 is known as the “guardian of the genome” thanks to its capacity to respond to outside stresses, which promotes transient or permanent cycle arrest and apoptosis, following different stress factors including hypoxia, DNA impairment, oxidative stress, hyperproliferative signals, nutrient shortage [19,20,21]. p53 supports tumor suppression through its roles as transcription factor and mitochondrial membrane permeabilization (to trigger apoptosis) and, indeed, the most investigated biological activity of p53 is its transcriptional activator role [17].

### 1.3. Transcriptional Role of p53, Relevant Mutations and the Mutant p53 GOF

In common with other transcription factors, p53 is composed of three distinct domains, which are responsible for oligomerization, transcriptional activation and sequence-specific DNA-binding [22,23]. Although the carboxy-terminal domain has been shown to play a pivotal role for the tetramerization of p53 monomers—which, in turn, triggers transcriptional activation—approximately 30% of TP53 mutations in human malignancies occur in six “hotspot” amino acid residues within the DNA-binding domain (R175, G245, R248, R249, R273 and R282). Such alterations in malignancies highlight the crucial role of p53 as a transcription factor in tumor suppression [24,25]. Together with the well-established tumorigenic promotion via loss of p53-wt function, the retained p53-mut is also thought to promote tumorigenesis via GOF properties and a dominant negative effect on the p53-wt protein [19,26]. After showing malignant characteristics such as increased survival and invasiveness, a poor differentiation rate and proliferation in preliminary cell culture studies, the GOF of tumor-derived p53-mut was corroborated by knock-in mice studies. In these studies, different mice strains harboring equivalent human tumor p53-mut (e.g., p53^273H^, p53^R248Q^, p53^R248W^, p53^175H^ and p53^G245S^) reported a shorter tumor latency, a wider tumor spectrum and a higher rate of metastasis, supporting the concept that p53-mut actively sustains cancer development and progression [13,17,27,28,29,30]. p53 GOF could be attributable to an increase in the interaction with other transcriptional factors such as the vitamin D receptor (VDR), the nuclear factor Υ (NFΥ) and p63 and p73 [31,32], resulting in the alteration or inhibition of the DNA-binding capacity of these transcription factors [33,34]. Lastly, p53-mut has been observed to interact with the Nijmegen breakage syndrome protein 1 (NBS1), Meiotic Recombination 11 (MRE11) and DNA repair protein RAD50 (RAD50), to make a complex (NBS1-MRE11-RAD50) and impair the DNA damage signalling [35,36].

### 1.4. p53 Degradation and Accumulation

The p53 transcription factor is part of an autoregulatory loop where p53 protein promotes the expression of Mouse double minute 2 homolog (MDM2) protein, which, in turn, polyubiquitinates the p53 protein and supports its degradation [10,37]. Yue et al. showed that BAG family molecular chaperon regulator 2 (BAG2) binds to p53-mut and translocates it into the nucleus to inhibit the interaction between MDM2 and p53, thus blocking the MDM2-mediated ubiquitination and degradation of p53. They further reported that the DBD of the p53-mut and the BAG domain of the BAG2 are essential for the BAG2-p53-mut interaction. Noteworthily, different models expressing p53-mut, that lacked the main sites for p53 ubiquitination by MDM2, still showed a normal half-life and stabilization for p53 under stress conditions [38,39]. In fact, there are alternative pathways involved in the degradation of p53. Benchimol et al. showed, for the first time, that MDM2 is not the only E-3 ligase capable of controlling p53 ubiquitination [40]. There are many different E3 ligases targeting p53 for ubiquitin-mediated degradation and their description is out of the scope of this review. For a more thorough description of such molecules involved in p53 degradation we refer to tailored reviews [41,42]. Briefly, the following are E-3 ligases involved in p53 ubiquitin-mediated degradation that mediate the K48-linked polyubiquitination of p53, marking the protein for proteosomal degradation: MDM2, Fogol Caspase Recruitment Domain-Containing protein 16 (COP1) and Ring Finger And CHY Zinc Finger Domain Containing 1 (Pirh2), as well as Tripartite Motif Containing 24 (TRIM24) [43], TOP1 Binding Arginine/Serine Rich Protein (TOPORS) [44], WW Domain Containing E3 Ubiquitin Protein Ligase 1 (WWP1) [45], ADP ribosylation factor–Protein oikybromo-1 (ARF-BP1) [46], Ubiquitin-conjugation enzyme E2 13 (Ubc13) [47], Human Herpes Virus Infected Cell Polupeptide 0 (ICP0) [48], Cell Division Cycle And Apoptosis Regulator 1 (CARP1/2) [49], Cullin 7 (CUL7) [50], Synoviolin (SYVN) [51], E4F Transcription Factor 1 (E4F1) [52], STIP1 Homology And U-Box Containing Protein 1 (CHIP) [53], E4orf6 and E1B55K [54], MSL Complex Subunit 2 (MSL2) [55], F-Box Protein 42 (JFK) [56] and Makorin Ring Finger Protein 1 (MKRN1) [56]. Notably, mono- or K63-linked polyubiquitination is linked to other functions besides ubiquitination, such as nuclear and cytosolic localization, alterations of p53 transcriptional levels. Nuclear export is induced by conjugating with ubiquitin by MDM2, MSL2 and WWP1; ICP0 was shown to accumulate ubiquitin p53 at nuclear foci; E4F1, was shown to active cell-cycle arrest at G0/G1 after mono-di or tri- ubiquitinating p53; CUL7 was shown to mono- or deubiquitinate p53 to repress p53 transcriptional activity by unknown mechanisms [45,48,50,52,55]. Therefore, there are different mechanisms by E3 ligases to fine-tune the levels of p53 in cells. However, none of these E3 ligases has been studied in detail as much as MDM2. p53 mutated proteins do not activate the expression of MDM2. In consequence, such mechanisms of degradation do not occur in p53-mut where MDM2 levels are very low [57]. Therefore, this has been postulated as an explanation of why only p53-mut bearing patients result in the formation of p53 antibodies (p53-Abs).

The accumulation of p53 can be triggered by many mechanisms, such as stress signals, DNA damage, nucleotide deprivation, DNA damage, mitogenic or oncogenic activation viral infection, heat shock proteins like HSP70/HSP40/HSP90, which, in cancer cells, form a multi-chaperone complex around p53-mut facilitating the unfolding of the p53-mut and its spontaneous folding to another conformation with different energy minimum [58,59,60] (Figure 1). In addition, the activity of p53 can be further enhanced by post-translational modifications working as positive or negative regulators [6,58,61,62].

## 2. p53 Role in Immunology as an Antigen Presented by Cancer Cells

p53 has been massively investigated over the last 40 years in the field of cancer biology; however, if, in this research thread, p53 had been studied by key players from within the field of immunology, the results may well have been quite different. Indeed, if the immune system recognizes tumor antigens and more than 50% of human cancers shows p53-mut, it is no wonder that these two approaches can be joint to achieve a communal goal.

It is worth mentioning that there are immune-related cellular mechanisms that p53 can trigger that are not available when p53 is mutated. Briefly, the following immune responses are influenced by the p53 status in tumors: (1) perturbations in the antigen presentation machinery, reducing immune escape [63]; (2) downregulation of natural killer group membrane D (NKG2D), which leads to cancer cells immune evasion [64]; (3) the regulation of immunosuppressive molecule programmed death ligand 1 (PD-L1) [65]; (4) the regulation of cytokines and chemokines [66]. Additionally, Major Histocompatibility Complex I (MHC-I)-bound peptide translocation into the endoplasmic reticulum requires antigen processing 1 and 2 (TAP1 and TAP2). When directly activated, as in response to DNA damage, p53 activates TAP1, therefore increasing the MHC-1 peptide complexes in tumors [67]. Such a mechanism is not available in p53-mut cells. A stable complex consists of MHC, a peptide and a T cell receptor. After being fully activated, effector T cells upregulate co-inhibitory receptors, such as PD-L1, to keep the immune system in check. Cancer cells bypass such inhibition by the expression of co-inhibitor ligands (e.g., PD-L1) to block the T cell activity directed against them [68]. In line with this, in lung cancer, p53 has been found correlated with PD-L1 expression, which could help identify patients who would best respond to PD-L1 [69]. Toll-like receptors (TLRs) are a family of conserved receptors altering the immune system of the presence of foreign agents, both intra- and extra- cellularly [70]. p53 transcriptionally targets TLR3 and TLR9, activating their expression. Such p53-depedent activation provokes apoptosis. Again, this response is lost in most cancer cells that express a p53-mut [71,72]. These cytokines could attract pro-tumorigenic immune molecules to the tumor site and foster tumor formation [73].

Given this background, separate studies observed that 10% of women diagnosed with breast cancer and approximately 30% of patients diagnosed with a wide range of cancers had detectable and quite specific antibodies against p53, correlated with cancers carrying p53 missense mutations [10]. As soon as the p53 protein epitopes (regions where p53 antibodies are binding) maps were unveiled, the N- and C-termini of p53 were reported to be the preferred regions to interact with antibodies. However, these are not the sites where missense mutations in the vast majority of p53-mut take place. Many p53-mut proteins are unable to bind to DNA and exert transcription factor properties like their native counterpart. In consequence, missense mutations on the TP53 gene do not transactivate the MDM2 gene and the autoregulatory loop fails, resulting in higher levels of p53-mut protein in cancerous cells [37,74]. Increased levels of a cellular protein can trigger the immune response and explain the antibody production against the p53 proteins [10].

p53-mut proteins have a longer half-life than p53-wt and therefore stay longer in cancer cells. In fact, it has been shown that inactive p53-mut have a half-life of several hours compared with 20 min for p53-wt. In consequence, the p53-mut accumulates in the nucleus of neoplasmic cells [75]. The accumulation of the p53-mut protein could, in turn, induce circulating p53 antibodies (p53-Abs) in cancer patients [76,77]. p53-Abs production appears to be an early event in some cancers. High levels of p53-Abs have been detected in patients with premalignant and malignant lesions, which could imply that they could be used as an early diagnostic tool for cancer detection before its occurrence [78]. Additionally, saliva p53-Abs have been also discovered and they could further provide an easier and less-invasive method to verify the prognostic utility of these antibodies [79].

The scope of this review is to provide an update on the current knowledge of the expression of serum p53-Abs (s-p53-Abs) and their prognostic value in cancer patients. For a more comprehensive review on the different molecular interactions between p53-wt and immune molecules, please refer to Blagih et al. [80].

## 3. Mechanisms of Regulation of p53 Wild-Types and Mutants

The p53 protein influences the innate immune system. Since evolutionary conserved viruses have developed a mechanism for p53 inactivation, this protein is not only the “guardian of the genome”, but it is also an essential part of the “innate immune response”, defending the cells against foreign viral offences.

The loss of p53 function is responsible for deep changes in the secretion of both chemokines and cytokines, which largely modify the immune environment [81,82,83,84]. In prostate, ovarian and breast cancers, p53 loss has been shown to recruit tumor-supporting myeloid cells [82,83,84]. The p53 loss in breast cancer within the tumor promoted the dysregulation of WNT signalling pathway and an increase in neutrophil circulation that supported the growth of cancer and spread into different organs [82]. In addition to the loss of p53, the loss of Phosphatase and Tensin Homolog (PTEN) enhanced the secretion of C-X-C Motif Chemokine Ligand 17 (CXCL17), which, in turn, recruited tumor-associated polymorphic myeloid-derived suppressor cells. Bezzi *et al.* demonstrated in such model that blocking the CXCR2 receptor diminishes tumor growth [84]. p53 loss induced an increase in tumor-associated macrophages (TAMs) in lung, ovarian, pancreatic and skin cancers [83,85]. The inhibition of the colony stimulatory factor 1 receptor (CSF1R) blocked the differentiation of macrophages, their survival and reduced TAMs in p53^R172H^ or p53^flox/flox^ and reduced xenografted pancreatic tumor growth [86,87].

Recently, Blagih et al. further demonstrated that p53 deletion promoted the recruitment of ICluster of Differentiation Molecule 11B (CD11b+) myeloid cells. From this recent study, it appears that the deletion of p53 and activation of KRAS Proto-Onogene, GTPase (KRAS) collaborated in the promotion of immune tolerance. In this pre-clinical study, it was also shown that the ability of p53-ko cancer to suppress the immune system was overcome by CSF1R neutralization and Tregs depletion [66].

Moreover, once the KRAS pathway is activated, it is sensed by p53 and the cells go through senescence, an often irreversible process that blocks cellular division. KRAS activates Ets gene transcription, which then leads to the transcription of tumor suppressor ARF. ARF protein is known to bind to MDM2 E-3 ubiquitin ligase. This interaction, in turn, blocks the activity of the ligase, which is needed to degrade p53. In consequence, the level of the p53 protein in the cells increases dramatically. The higher levels of p53 protein lead to an increase in other tumor-suppressing transcriptional pathways, such as Plasminogen Activator Inhibitor 1 (PAI-1), Promyelocytic Leukemia (PML), p21, and mir-34r. Such pathways inhibit the cell cycle and promote senescence. These senescent cells produce a high number of interleukins and cytokines, attracting thereby CD-8 killer T cells, NK cells, macrophages and monocytes, which kill senescent cells [88]. In this scenario, p53 activation with KRAS mutation works as an immune system defence mechanism to eradicate the senescent cells from the body. It is well known that, with age, the immune system becomes worse. It is also known that aging is related to an increase in senescent cells [89]. This could imply that aging is controlled by the innate immune response governed by p53 during the aging process [90,91]. It is intriguing how p53 and KRAS function by killing senescent cells through the innate immune system. Furthermore, blocking p53-MDM2 interaction could be a method to further exploit in order to push the virally infected cancer cells to death. Such therapeutic drugs could be useful to treat not only cancers caused by viruses, such as SV-40 or papillomaviruses, but also life-threatening diseases, and could play a role in the future of the construction of novel treatments and also against the current SARS CoV-2 pandemic, thereby blocking the powerful pro-inflammatory cytokine storm (e.g., the production of IL-6, IL-12 and TFN-α). The cytokine storm is considered to be the cause of progression of COVID-19 patients to a more severe, critical pneumonia and multi-organ dysfunction, which becomes worse with the advancement of age [92]. A drug such as Nutlin-3 blocks MDM2-driven p53-wt degradation. Therefore, virally infected cells would go through p53-wt mediated apoptosis.

The figure summarized how viral infection, genotoxic, proteotoxic and oxidative stress regulate the stability of p53-wt or p53-mut. The wild-type p53 protein is degraded through MDM2-mediated ubiquitination. This process is inhibited in case p53 is mutated. Nutlin-3 is a drug that has been investigated to block p53 wild-type degradation. Current p53 antibodies might not make a clear distinction between the p53 wild-type or p53 mutant. Such antibodies could become therapeutically efficient in blocking the oncogenic role of p53-mut.

## 4. p53-Abs Measurements and Specificity

There are several methods for the measurement of p53-Abs from specimens, such as ELISA [93,94], Western blot or immunoprecipitation [95,96], possibly explaining why results are different in the literature regarding the frequency of p53-Abs. Additionally, there are different types of epitopes recognized by these assays. p53-Abs usually recognizes immunodominant epitopes found in the COOH and NH2 termini of p53. Therefore, it is crucial to have valid systems that recognize the entire p53 protein, looking at both sides [97].

The structure of p53 has already been established by X-ray crystallography. The protein has a central region made of 102–292 amino acids [98]. This includes two β-sheet antiparallel motifs made of four and five β-strands, respectively. Moreover, these strands form a structure that keeps together the protein: (1) Loop-sheet-helix (LSH) with an α-helix, three β-strands and the L1 loop; (2) the L2 loop with a smaller helix; (3) the L3 loop made of turns. The regions involved in the DNA interaction are the LSH motif and L3 helix. According to Soussi *et al.*, these two loops stabilize the protein structure by binding with each other by a zinc atom connected to Cys176 and His179 on the L2 loop and Cys278 and Cys242 on the L3 loop [75,99]. Additionally, the central region of p53-wt is held in a compact conformation by two antiparallel β-sheets. This specific conformation is hydrophobic and poorly immunogenic [99]. Some key epitope residues used by anti-p53 Abs correspond to those needed for the MDM2 interaction with the NH2 terminus of p53 [7].

p53 antibodies could be specific to at least one conformation of p53-mut. Interestingly, there are some publications showing that p53-Abs are able to recognize not only the p53-mut but also the p53-wt [100,101,102]. The epitopes of human p53 have been precisely mapped [103]. From an analysis of more than 200 sera, it was possible to evince that the immune-dominant epitopes were in the COOH terminus of human p53, whereas only very few antibodies recognized the human p53 central region of the protein [101,103,104,105].

## 5. p53-Abs in Cancer Patients

The first population study looking at s-p53-Abs produced by human cancer patients relates back to 1982, where it was demonstrated that p53-Abs were found in 10% of the sera of female breast cancer patients [96]. More studies on various cancers came afterwards, proving that, in about 30% of different types of cancers, p53-Abs were detected [75].

The s-p53-Abs have been found in patients with head and neck cancer, the fluids of ovarian cancer ascites, pancreatic, colon and lung tumor pleural effusions, and saliva from oral cancer patients [75,79,106,107,108]. As expected, there is a high correlation between the s-p53-Abs and the frequency of p53 gene alterations in the tumors [79,93,100,107,109,110]. In mesothelioma and prostate cases where p53-mut were low, the presence of s-p53-Abs was low [93]. Tavassoli et al. showed that p53 was detected in both saliva and sera of 29 oral SCC patients [79]. Angelopoulou et al. analysed the fluid of 96 ascites from women with primary ovarian cancer for s-p53-Abs. They showed that autoantibodies were present in both the serum and ascites of 6/30 patients; of them, 22 were negative for auto-antibodies in either serum or ascites, and two had them only in the serum [107].

Soussi et al. made a meta-analysis of 18 clinical studies, examining 9489 patients with cancer, and discovered that the s-p53-Abs were a marker for patients with neoplasia (*p* < 10^−4^) [75]. There was a significant correlation between s-p53-Abs and p53 mutations and various types of cancer. Interestingly, cancers that did not have p53-mut, specifically testicular carcinoma [108,111], hepatoma [112] and melanoma [113,114], were also negative for p53-Abs. On the other hand, most cancers with high rate of p53-mut have also a high frequency of s-p53-Abs, except for glioma, with a high rate of p53-mut [115] and a low rate of s-p53-Abs [115,116]. In a glioblastoma cohort of 60 patients, 24 of them had p53-mut, but none of them had s-p53-Abs. This could be due to the lack of the p53 immune response in the brain (an immunoprivileged organ), preventing the production of s-p53-Abs.

Roughly, about 20–40% of p53-mut patients had the s-p53-Abs in their sera. It is worth mentioning that, despite similar tumors, p53-mut and its accumulation, it is not a rule that all these patients are positive for s-p53-Abs [117,118,119,120]. There must be other factors that come into play in the production of s-p53-Abs that future research could unravel.

## 6. Prognostic Value of p53-Abs in Cancer Patients

Many studies have been conducted to evaluate the clinical utility of s-p53-Abs. In Breast Cancer (BC), there are several studies showing that s-p53-Abs correlated with higher tumor grades [94,96,101,121,122]. Lenner et al. showed, in 353 BC patients, that the presence of a high concentration of s-p53-Abs correlated significantly with worse survival (*p* = 0.003) [123]. Likewise, Peyrat et al. showed, in 165 BC patients, that overall survival was worse in patients with s-p53-Abs (*p* < 0.0005) [122].

Notably, Willsher et al., in 82 BC patients, did not find any correlation [124]. Porzsolt et al., in 50 BC patients, showed that p53-Abs were present in BC patients with a good prognosis [125]. On the contrary, Generali et al. showed an association between the absence of ER expression and p53 expression, supporting its negative predictive role. Moreover, hypoxia-inducible factor-1α (HIF-1α) is directly linked to the p53-mut in BC (noteworthily, routine immunohistochemical procedures usually detect the mutated form); this leads to the inhibition of tumor apoptosis with a negative impact on treatment response in BC [126]. In colorectal cancer (CRC), a few initial studies demonstrated that s-p53-Abs correlated with worse survival [109,110,127]. In fact, Kressner et al. showed, in 184 CRC patients, that s-p53-Abs correlated with shorter survival. However, the latter data did not reach the same statistical significance when Duke’s stage was taken into consideration in the analysis model [127]. Houbiers *et al.* conducted a post-operative study in 255 CRC patients using ELISA. The group showed that 25.5% of the patients were s-p53-Abs-positive. The presence of such antibodies significantly correlated with prognostic factors, like tumor shape, histological grade, angiogenesis and Quetelet Index (*p* = 0.02, *p* = 0.04, *p* = 0.02 and *p* = 0.01, respectively). Moreover, in 64 of the CRC patients, who were at stage A or B1, s-p53-Abs positivity significantly correlated with decreased overall survival and disease-free survival (*p* = 0.04 for both) [110]. On the contrary, more recently, Kunizaki et al., in 170 CRC patients, observed that serum s-p53-Abs did not correlate with overall survival [128]. Angelopoulou et al., in 229 CRC patients, showed that s-p53-Abs correlated with disease stage and progression. Although their data failed to reach statistical significance, they postulated an interesting hypothesis that s-p53Abs could be used to monitor CRC disease progression [109]. The result could be due to the specific p53 mutation (c.742C > T p.R248W) that the serum p53 antibody recognized. Perhaps testing for another serum p53 antibody—recognizing another p53 mutation—could be of prognostic value in CRC. However, Tokunaga et al., with 244 CRC patients, more recently observed that serum s-p53-Abs were not able to predict patients’ prognoses (*p* = 0.788) [129]. Their serum p53 antibody recognized wild-type p53.

Furthermore, in lung cancer, there are controversies regarding the clinical utility of s-p53-Abs. In fact, in non-small-cell lung carcinoma (NSCLC) the antibodies are associated with worse survival [130,131,132]. Moreover, Mack et al. detected, in 180 lung cancer patients, that there was a significant correlation between p53-Abs positivity and a shorter survival of NSCLC patients (*p* = 0.01) [133]. More recently, Mattioni et al. conducted a study of 201 NSCLC patients. In fact, the authors demonstrated that NSCLC patients with lower levels of s-p53-Abs survived significantly longer than patients with higher levels of s-p53-Abs (*p* = 0.049). Along the same lines, the same publication additionally showed that patients with squamous cell carcinoma (excluding adenocarcinoma) with lower levels of s-p53-Abs survived significantly longer compared to patients with higher levels of s-p53Abs (*p* = 0.044) [108].

In small-cell lung carcinoma (SCLC), Zalcman et al. observed, in 97 SCLC patients, that patients with s-p53-Abs had a worse survival rate compared to patients without s-p53-Abs (*p* = 0.014) [134]. Other groups corroborated such findings. On the contrary, Murray et al. showed, in 231 SCLC patients, that those patients with elevated s-p53-abs levels had a better survival compared to patients with lower levels of the antibody. In fact, by setting, through ELISA, a higher threshold for p53 positivity by choosing a score up to five, the authors observed that p53-Abs positive patients had a median survival of 11 months vs. 8 months of the patients that did not have the antibody. p53Abs appeared to be an independent prognostic factor in this study (*p* = 0.02) [77].

Another series of surprising data come from oral cancer. There were two initial studies proving that s-p53-Abs expression correlated with worse survival of patients [103,104]. Bourhis et al., in 80 oral cancer patients who were s-p53-Abs positive, evinced that s-p53-Abs expression correlated with a higher risk of tumor relapse and death (*p* = 0.003 and *p* = 0.03, respectively) [135]. Werner et al., in 149 patients, conducted a clinical study made of head and neck cancer patients who received surgery and radiation therapy. In line with what was observed by the former group, the latter group detected that the 17 seropositive for s-p53-Abs patients (44.7%) failed to respond to the therapy, whereas only eight of the seronegative patients for the s-p53-Abs failed to respond to therapy (21.1%) [136]. Later, along the same lines as the previous two groups, Gottschlich et al., with 109 head and neck cancer patients, observed that s-p53-abs correlated with worse outcomes, although the authors did not show any statistical significance in their data [137]. Successively, Shimada et al. detected, in 258 oesophageal cancer patients, that s-p53-Abs (*p* < 0.001; HR: 10.62; 95%CI:0.76-40.00) and serum C-reactive protein concentration (S-CRP) were independent prognostic factors [138]. To corroborate these findings, later, Sainger et al. observed that the s-p53-Abs positivity correlated with lymph node metastasis, advanced disease and well-differentiated tumors [139]. More recently, Kunizaki et al. attested, in 133 patients with esophageal squamous cell carcinoma, that the high levels of both s-p53Ab and squamous cell adenocarcinoma antigens in patients correlated with significantly lower survival compared to patients with elevated levels of only one or neither of these factors (*p* = 0.009) [140]. Furthermore, s-p53-Abs positivity strongly correlated with poor outcomes from treatment in in the 60 oral precancerous treated patients [139].

An updated list of all the clinical studies evincing the prognostic value of s-p53-Abs in all the cancer types is summarized in Table 1.

## 7. Discussions and Future Directions

The role of p53-Abs is becoming increasingly important in an era where immune-oncology is advancing at a great speed. Firstly, it is known that infection by SV40 and any other virus stresses the cell and that the infecting viruses avoid cell death by elaborating the following mechanisms: they mediate the degradation of p53-wt protein and they inhibit the Rb protein, both of which are needed to cause apopotosis and cell cycle arrest. Secondly, in many cancer cells, from viral and non-viral origin, p53-mut is expressed at very high levels. The p53-mut is unable to bind to DNA to promote the transcription of MDM2 and other p53 transactivational targets and it is therefore not regulated by the same autoregulatory loop of p53-wt. As a direct consequence, mutant p53 proteins become often stable and accumulate at very high levels inside tumor cells. As an indirect consequence, the accumulated p53-mut serves as an antigen. Finally, these p53-mut antigens drive the adaptive immune system to recognize such antigens and produce antibodies that may kill the cancer cells with p53-mut or normal cells with p53-wt. Notably, the studies investigating the prognostic role of p53-Abs still remain controversial, and this appears to be tissue specific. Most interestingly, we have reported here a full updated list of studies on the prognostic value of s-p53-Abs in cancer from the published literature (Table 1). Except in a few cases, the vast majority of these studies use p53-wt as the antigen to detect s-p53-Abs in cancer patients and it is unclear whether these s-p53-Abs can recognize p53-mut at all. In an historic epoch that has been contemplating the advancement of targeted therapies to meet specific therapeutic needs of cancer patients, it would be of paramount importance to know whether the ELISA assays, designed for the recognition of s-p53-Abs, can recognize p53-mut. This review tries to summarize, to the best of our knowledge, this notion in Table 1. Having methods to detect specific p53-Abs recognizing specific p53-mut could become pivotal in future studies trying to better understand their prognostic value. Moreover, antibodies against specific p53-mut could be directly used for future therapies, which is something p53 biologists would like to further investigate. Sabapathy and colleagues have put this idea into practice by developing mouse monoclonal antibodies against p53 hotspot mutants, R175H, R248Q, and R273H [154], but their therapeutic efficacy has not yet been reported. A recent study showed that a T cell receptor-like antibody, initially made for a p53-wt antigen, was capable of discriminating between p53-mut and p53-wt-expressing cancer cells by killing p53-mut cells more effectively than p53-wt cells in vitro and inhibiting the growth of p53-mut xenograft tumors. This is likely due to the differences in wt- and mut-p53 accumulation and degradation, generating the peptide to be bound by the antibody [155].

Notably, the expression of PD-L1 and p53 correlated in various types of cancers [156,157,158,159,160]. All these groups used immunohistochemistry, which detected p53-mut. Cortez et al. showed that PD-L1 is a direct target of microRNA-34 (mir-34), using Western blotting and a luciferase assay. Using a p53 knocked down mouse model, they further showed that such regulation was dependent on the presence of p53-mut. p53-mut downmodulates the expression of PD-L1 through mir-34 [156]. For such reasons, it would also be interesting in future experiments to investigate the efficacy of p53-Abs recognizing p53-mut in combination with checkpoint inhibitors against PD-L1.

In conclusion, there are s-p53-Abs in a significant portion of cancer patients (mostly observed by ELISA) and only in a small fraction of these studies does the presence of s-p53-Abs predict better survival. That was the case for two studies of BC, two studies of NSCLC, and one study of SCLC. On the contrary, more studies showed that the presence of s-p53-Abs predicted worse survival. Such a result was mostly predominant in head and neck cancers—in five studies—followed by CRC—in four studies—and NSCLC (in three studies). Virtually all s-p53-Abs are detected using p53-wt as an antigen. One thing we do not currently know, but hope to know, is whether an antibody targeting a specific p53-mut will be effective in therapeutics against the cancer with a specific mutation in the p53 gene. Although our review showed that, in most of cases the presence of s-p53-Abs correlated with worse survival, it would be interesting to investigate whether p53 antibodies recognize p53 hotspot mutations, including c.659A > G (p.Y220C) and c.733G > A (p.G245S), which have been uniquely discovered in human ovarian cancers [161]. Through targeted therapies, it could be possible to treat p53-mut in cancer cells, while sparing normal cells, with the aim of improving efficacy and reducing toxicity to patients, with respect to conventional therapies. As a matter of fact, s-p53-Abs from cancer patients may target both p53-wt and p53-mut. Novel antibodies targeting p53-mut, but not p53-wt, should be pursued in pre- and clinical trials.

## Figures and Tables

**Figure 1 ijms-21-04087-f001:**
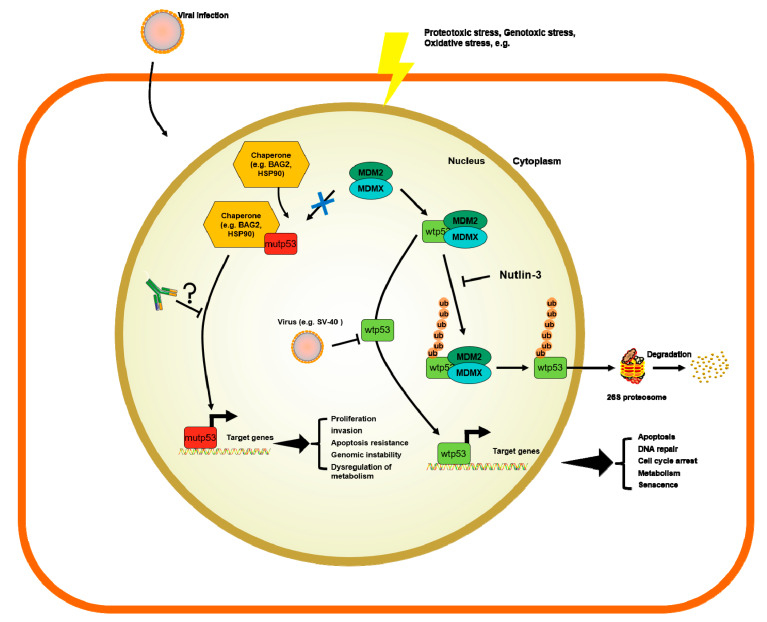
Mechanisms of regulation of wild-type p53 (p53-wt) and p53 mutants (p53-mut) and the potential role of p53 antibodies (p53-Abs).

**Table 1 ijms-21-04087-t001:** Clinical studies investigating the prognostic value of serum p53 antibodies (s-p53-Abs) in cancer.

Study	Methods and Patients	Prognostic or Predictive Outcomes	Reference
Kunizaki et al., 2018	S-p53Ab, SCC-Ag, CEA Antibody for p53-wt 133 esophageal squamous cell carcinoma patients	The presence of both S-p53Ab and SCC-Ag in patients correlated with significantly lower survival compared to patients with elevated and patients with elevated levels of only one or neither of these factors (*p* = 0.009).	[140]
Kunizaki et al., 2017	S-p53Ab Detected with anti-p53 detection kit MESACUP anti-p53 Test Antibody for p53-wt 208 GC patients	Did not observe any significant correlation between S-p53Ab in GC and overall survival (hazard ratio(HR) = 2.052; 95% confidence interval(CI) = 0.891–4.726; *p* = 0.091). Conversely, Cox regression analysis revealed that a high level of CA19-9 was an independent prognostic factor for GC (hazard ratio(HR) = 3.864; 95% confidence interval(CI) = 1.248–11.959; *p* = 0.019).	[141]
Tokunaga et al., 2017	CEA, CA19-9, S-P53Ab Antibody for p53-wt 244 CRC patients	S-P53Ab had no power to predict the prognosis (*p* = 0.786). Combined CEA and CA19-9 positivity was an exclusive independent prognostic factor (*p* = 0.034).	[129]
Kunizaki et al., 2016	S-p53Ab, CEA ELISA. Antibody for p53-wt 170 CRC patients	Positivity for s-p53Ab in CRC did not correlate with overall survival. Kaplan-Meier analysis revealed significant differences between patients with elevated s-p53Ab and CEA and those with elevated levels of either one or neither of these factors (*p* < 0.001).	[128]
Mattioni et al., 2015	s-p53-Abs ELISA. Antibodies for p53-wt Direct Sequencing was used to detect mutations: Mutant and Wild-type antibodies: nine patients with p53-wt and three patients with p53 deletions 201 NSCLC patients	Patients with lower levels of p53Abs survived significantly longer than patients with higher levels of p53Abs (*p* = 0.049).	[108]
Anderson et al., 2010	s-p53Ab ELISA. Antibodies for p53-wt Invasive serous OC (*n* = 60), non-serous ovarian cancers (*n* = 30), and women with benign disease (*n* = 30).	p53-Ab did not significantly improve the detection of cases [area under the curve (AUC), 0.69] or the discrimination of benign versus malignant disease (AUC, 0.64) compared with CA 125 (AUC, 0.99) or HE4 (AUC, 0.98). In multivariate analysis among cases, p53-AAb correlated only with a family history of breast cancer (*p* = 0.01).	[142]
Atta et al., 2008	s-p53Ab ELISA. Antibodies for p53-wt. 41 HCC, 26 Liver cirrhosis, 29 healthy controls	Our results revealed that anti-p53 has a positive significant correlation with AFP (*p* = 0.002), severity of liver disease [Child Pugh score (*p* = 0.02) and MELD score (*p* = 0.0003)], tumor size (*p* < 0.0001), tumor number (*p* = 0.003) and tumor staging systems [Okuda (*p* = 0.04), CLIP (*p* = 0.006) and Tokyo (*p* < 0.0001)]. Moreover, our results revealed that s-p53-Abs had a significant association with overall survival of patients with HCC (*p* = 0.019) with a shorter survival time in anti-p53 positive status patients and with higher s-p53-Abs levels within 19 months follow up.	[143]
Mattioni et al., 2007	S-p53-Abs Levels of p53-mut were determined with a selective, quantitative ELISA kit (Cambridge, Oncogene, USA) 111 GC patients 64 healthy donors	The survival time of serum-positive patients was significantly longer than that of patients with low/negative serum levels, with a survival rate of 41.2% and 14.9%, respectively, over 48 months (*p* < 0.05).	[144]
Lawniczak et al., 2007	S-p53-Abs ELISA 71 GC patients	The presence of p53-Abs was connected with intestinal tumor type (*p* < 0.05) and older age (*p* = 0.0035).	[137]
Akere et al., 2007	S-p53-Abs ELISA. Antibodies for p53-wt 41 HCC patients 45 controls	There is a low prevalence of serum anti-p53 in our study population, and this is commoner in men. It is also present in the control group and therefore may not be useful as a diagnostic tool in this study population.	[145]
Sainger et al., 2006	S-p53-Abs ELISA. Antibodies for p53-wt 60 oral precancerous patients, 75 untreated oral cancer patients, and 86 follow-up blood samples of the oral cancer patients. 55 healthy controls,	The s-p53-Abs positivity correlated with lymph node metastasis, advanced disease and well-differentiated tumors. Furthermore, p53-Abs positivity strongly correlated with poor outcome from treatment in in the 60 oral precancerous treated patients.	[139]
Goodell et al., 2006	S-p53-Abs ELISA. Antibodies against p53^K132Q^ (c.394A > C). 104 ovarian cancer patients	Patients with s-p53Abs recognizing the mutated protein showed a significantly higher survival compared to patients without antibody (*p* = 0.01).	[146]
Gumus et al., 2004	S-p53-Abs ELISA. Antibodies for p53-wt 76 urinary bladder cancer patients	There was an association between the presence of s-p53-Abs and tumor p53 gene overexpression (*p* = 0.001).	[147]
Shimada H et al., 2003	S-p53-Abs, C-reactive ELISA Antibodies for p53-wt 258 oesophageal cancer patients	s-p53-Abs (*p* < 0.001; HR: 10.62; 95%CI:.76–40.00) and S-CRP were independent prognostic factors.	[148]
Hødgall et al., 2002	S-p53-Abs ELISA Antibodies for p53-wt 193 OC patients 34 borderline OC 86 healthy controls	No significant associations were found between p53 AAb and clinical stage, age, histological subtype and radicality after primary surgery.	[149]
Parasole et al., 2001	S-p53-Abs ELISA. Antibodies for p53-wt 80 HCC patients	Anti-p53 was not useful as a prognostic factor.	[150]
Tangkijvanich et al., 2000	S-p53-Abs ELISA. Antibodies for p53-wt 121 HCC patients	There were no differences between groups with regard to age, sex, viral markers (HBsAg or anti-HCV), severity of liver disease and tumor advancement. The median survival rates for patients positive and negative for s-p53-Abs were 4.0 and 3.0 months, respectively (*p* = 0.443, by log-rank test).	[151]
Sitruk et al., 2000	S-p53-Abs ELISA 159 HCC patients	Detection of s-p53-Abs was significantly correlated with the presence of a multinodular or infiltrative tumor (*p* < 0.03).	[152]
Zalcman et al., 2000	S-p53-Abs ELISA. Antibodies for p53-wt 97 SCLC patients	Patients with limited-stage SCLC and p53-Ab had a median survival time of 10 months, whereas limited-stage SCLC patients without p53-Ab had a 17-month median survival time (*p* = 0.014).	[134]
Murray et al., 2000	S-p53-Abs ELISA. Antibodies for p53-wt 231 SCLC patients	High levels of p53-Abs correlated with worse survival compared to patients with lower levels of the antibodies (*p* = 0.02).	[77]
Gottschlich et al., 2000	S-p53-Abs ELISA. Antibodies for p53-wt 109 head and neck cancer patients	p53-seropositive for the p53-Abs patients showed a correlation with clinical outcome.	[138]
Mack et al., 2000	S-p53-Abs Immunofluorescence. Antibodies against p53 ^R273H^ (c.818G > A). 35 SCLC patients 99 NSCLC patients	There was no correlation between p53-Abs status in SCLC, but the presence of these antibodies and a significant correlation with shorter survival in NSCLC (*p* = 0.01).	[133]
Lenner et al., 1999	S-p53-Abs ELISA. Antibodies for p53-wt 353 BC patients	There was a significant negative correlation between presence of s-p53-Abs and survival (*p*= 0.003).	[123]
Kressner et al., 1998	S-p53-Abs ELISA. Antibodies for p53-wt 184 CRC patients	p53-Abs correlated with shorter survival (*p* = 0.02).	[127]
Werner et al., 1997	S-p53-Abs ELISA. Antibodies for p53-wt 143 oral cancer patients	the presence of the p53-Ab significantly correlated with more local tumor recurrences and deaths tumor compared to the other group of p53-Ab negative patients (*p* < 0.05).	[136]
Angelopoulou et al., 1997	anti-p53 antibodies ELISA. c.742C > T p.R248W 229 CRC patients	p53Abs did not significantly correlate with survival.	[109]
Bourhis et al., 1996	S-p53-Abs ELISA. Antibodies for p53-wt. They previously showed a correlation between their ELISA p53-wt antibodies with the presence of p53-mut gene [115]. 90 oral cancer patients	p53-Abs expression correlated with a higher risk of tumor relapse and death (*p* = 0.003 and *p* = 0.03, respectively).	[135]
Willsher et al., 1996	S-p53-Abs ELISA Antibodies for soluble p53 with 132 amino acids deletions from N-terminus. 82 BC patients	Did not find any correlation.	[124]
Peyrat et al., 1995	S-p53-Abs ELISA. Antibodies for p53-wt. 165 BC patients	Overall survival was significantly worse in patients with s-p53-Abs compared to patients without the s-p53-Abs (*p* < 0.0005).	[122]
Houbiers et al., 1995	S-p53-Abs ELISA. Antibodies for p53-wt 255 CRC patients	Overall survival and Disease Free Survival were significantly worse in patients with s-p53-Abs compared to patients without the s-p53-Abs (*p* = 0.04 for both).	[110]
Porzsolt et al., 1994	S-p53-Abs ELISA. Antibodies against p53^R273H^ (c.818G > A). 50 BC patients	s-p53-Abs were higher in BC patients with high risk vs. patients with low risk. The difference was not statistically significant (*p* = 0.15).	[125]
Volkmann et al., 1993	S-p53-Abs ELISA. Antibody PAb 1801 against p53-wt and p53-mut. Epitope amino acids 32–79 80 BC patients	s-p53-Abs correlated with better prognosis compared to patients without the antibodies (*p* < 0.00003).	[153]

Abbreviations: breast cancer (BC); hepatocellular carcinoma (HCC); gastric cancer (GC); serum p53 antibodies (s-p53-Abs); small-cell lung carcinoma (SCLC); p53 wild-type (p53-wt); p53 mutant (p53-mut); non-small-cell lung carcinoma (NSCLC).

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
