# Peer review of "Mutant p53 as an Antigen in Cancer Immunotherapy"

_ijms, 2020, doi:10.3390/ijms21114087_

Round 1
Reviewer 1 Report
The review by Dr. Sobhani and colleagues covers an important, still unresolved and timely topic on the role, implications and opportunities offered by the study of antibodies directed against p53 antigens or neoantigens in cancer.
The review covers a very high amount of literature, but, unfortunately, it does not fully deliver its aim, due in part to lack of focus, and in part to some sections or sentences that are not easy to read, at least in my opinion.
Also, there are some potential contradictions between the main text, table 1 and the conclusive paragraph.
I would suggest the Authors to improve the review by sharpening its focus on the topic declared in the title, and giving more emphasis to the data in Table 1. The table itself should be reorganized to highlight more effectively and concisely the tumor types being tested, the number of samples, the method used, and the results obtained in terms of prognostic or predictive outcome. Ideally, a meta-analysis combining the data of different studies could be considered.
I would then encourage the Authors to discuss in some more details what are in their expert opinion the reasons underlying the wide variation and apparent contradictory results reported in the literature, including limitations and shortcomings of the prevailing experimental approaches used.
Here is a list of typos, sentences to check, questions or criticism along the manuscript text
Line 24: carrying the exact mutant p53. Please rephrase.
Line 25: preclinical is misspelled
Line 43: higher than that. Please rephrase.
Line 52: this lack of functionality. Please clarify
Line 57: mutant”s”
Line 58: its cellular stress capability. Please rephrase.
Line 61: involving transcription inhibition. Please correct and rephrase
Line69: please correct/specify the information on the prevalence of the p53 mutation hostpots
Line80: might be again a transactivating role. Please rephrase.
Line97: read more reviews. Please rephrase.
Line111: please clarify why mutant p53 would not be degraded by MDM2
Line115: in could have happened in another way. Please rephrase.
Paragraph starting at line131: is the production of p53-Abs an early event in cancer? What about mutant p53 protein stability in the heterozygous state (before LOH)?
Paragraph starting at line136: these are relevant information, but mechanistically distinct from mutant p53 proteins triggering an immune response
The title of section 3 does not seem to match well with the section content.
Line184: myeloid is misspelled
Line204-212: check spelling “known”, “becomes”, “senescent”.
Line 208-215: please clarify how an approach inhibiting MDM2 can ameliorate the problem of pro-inflammatory-cytokine release.
Figure 1 could be rendered more informative and tailored to the main topic of the review
The specific topic of the review starts with section 4 -on page 6-. Consider to make the previous sections more concise.
Line234, 260 and elsewhere: please correct spelling “Soussi”.
Line247: please define “serum p53 antibody” when first introduced
Line251: “have”
Line276: please correct and clarify “hormone negative steroid hormones”
Line299: I understand that the hypothesis may be attractive, but is the data available supporting it? Please clarify if you consider that sample size or methodology issues may have biased the negative result.
Line303-305: please check the sentence.
Line309: can a more specific information be used to substitute the expression “strong level”
Line315: “corroborating what is the knowledge”?
Line318: “ adenocarcinoma”
Line 322: delete “of”
Line336: which there is capital S (p53-Ab). Also define SCC-Ag
Table 1. See above. I believe the table need to be restructured considerably.
Ideally a meta-analysis of the various studies (or at least of the tumor types for which more data is available) could be attempted. Or as a minimum, an attempt to interpret the opposite results that are frequent in the table in light of sample size or method used or other variables
Line355: why is the dominant-negative effect mentioned at this point? Is there conclusive evidence that in primary cancer heterozygous for mutant p53, the mutant protein is expressed at very high levels?
Line 357-358: the sentence could be modified and rendered less speculative adding to the discussion on specific mutant p53 neoepitopes could be expanded. Consider also citing the Rosenber/Deniger JCI paper.
Line374: please clarify in which sense mutant p53 act through mir-34. As it reads now the sentence does not seem to be correct.
The conclusive statement can be refined: is the fact the only a small fraction of studies shows that p53 antibodies predict better survival the main message of Table 1 and the review and how does this connect with the expected therapeutic benefit of antibodies targeting mutant p53?
Author Response
Peer-reviewer 1
The review by Dr. Sobhani and colleagues covers an important, still unresolved and timely topic on the role, implications and opportunities offered by the study of antibodies directed against p53 antigens or neoantigens in cancer.
Thank you.
The review covers a very high amount of literature, but, unfortunately, it does not fully deliver its aim, due in part to lack of focus, and in part to some sections or sentences that are not easy to read, at least in my opinion.
Also, there are some potential contradictions between the main text, table 1 and the conclusive paragraph.
I would suggest the Authors to improve the review by sharpening its focus on the topic declared in the title, and giving more emphasis to the data in Table 1. The table itself should be reorganized to highlight more effectively and concisely the tumor types being tested, the number of samples, the method used, and the results obtained in terms of prognostic or predictive outcome. Ideally, a meta-analysis combining the data of different studies could be considered.
I would then encourage the Authors to discuss in some more details what are in their expert opinion the reasons underlying the wide variation and apparent contradictory results reported in the literature, including limitations and shortcomings of the prevailing experimental approaches used.
Thank you for the suggestion. We have made the abovementioned corrections and simplified the form of the article to help the reader. Although a meta-analysis goes beyond the focus of the current paper, we have summarized the key finding of the prognostic value of s-p53-Abs at the final conclusion and fixed the article according to your and the other reviewer’s suggestions.
Here is a list of typos, sentences to check, questions or criticism along the manuscript text
Line 24: carrying the exact mutant p53. Please rephrase.
We have rephrased it.
Line 25: preclinical is misspelled
We have fixed it.
Line 43: higher than that. Please rephrase.
We have fixed it.
Line 52: this lack of functionality. Please clarify
We have clarified it.
Line 57: mutant”s”
We have fixed it.
Line 58: its cellular stress capability. Please rephrase.
We have rephrased it.
Line 61: involving transcription inhibition. Please correct and rephrase
We have corrected and rephrased it.
Line69: please correct/specify the information on the prevalence of the p53 mutation hostpots
We have fixed it.
Line80: might be again a transactivating role. Please rephrase.
We have rephrased it.
Line97: read more reviews. Please rephrase.
We have rephrased it.
Line111: please clarify why mutant p53 would not be degraded by MDM2
We have clarified it. We have added “p53 mutated proteins do not activate the expression of MDM2. As a consequence, such mechanisms of degradation do not occur in p53-mut where MDM2 levels are very low [56].”
Line115: in could have happened in another way. Please rephrase.
We have removed that phrase.
Paragraph starting at line131: is the production of p53-Abs an early event in cancer?
p53-Abs production appears to be an early event in some cancers. We have added such sentence.
What about mutant p53 protein stability in the heterozygous state (before LOH)?
Stability increases when p53 is mutated. We have added “In fact it has been shown that inactive p53-mut have a half-life of several hours compared with 20 minutes for p53-wt. As a consequence the p53-mut accumulated in the nucleous of neoplasmic cells[64].” There is no mention anywhere about the stability of p53 in the heterozygous state.
Paragraph starting at line136: these are relevant information, but mechanistically distinct from mutant p53 proteins triggering an immune response.
We have rearranged it into the initial part of section 2. To help clarify we have added “Worth to mention that there are immune-related cellular mechanisms that p53 can trigger that are not available when p53 is mutated”; and we have added numbers when briefly listing them.
The title of section 3 does not seem to match well with the section content.
We have changed it.
Line184: myeloid is misspelled
We have fixed it.
Line204-212: check spelling “known”, “becomes”, “senescent”.
We have fixed it.
Line 208-215: please clarify how an approach inhibiting MDM2 can ameliorate the problem of pro-inflammatory-cytokine release.
I have added the following to explain how it could be hypothetically possible: “A drug, such as Nutlin-3, blocks MDM2-driven p53-wt degradation. Therefore, virally infected cells would go through p53-wt mediated apoptosis. ”
Figure 1 could be rendered more informative and tailored to the main topic of the review
We have fixed it.
The specific topic of the review starts with section 4 -on page 6-. Consider to make the previous sections more concise.
We have removed the unnecessary parts. Thank you for the suggestion that we have considered. It looks great.
Line234, 260 and elsewhere: please correct spelling “Soussi”.
We have fixed it.
Line247: please define “serum p53 antibody” when first introduced
We have defined it when first mentioned.
Line251: “have”
We have fixed it.
Line276: please correct and clarify “hormone negative steroid hormones”
We have removed it.
Line299: I understand that the hypothesis may be attractive, but is the data available supporting it? Please clarify if you consider that sample size or methodology issues may have biased the negative result.
We have rephrased it to explain it better and we have added the following to explain what could have been the motive behind the negative result of such an interesting idea “The result could be due to the specific p53 mutation recognized by the specific mutation (c.742C>T p.R248W) that the serum p53 antibody recognized. Perhaps testing for another serum p53 antibody -recognizing another p53 mutation- could be of a prognostic value in CRC.”
Line303-305: please check the sentence.
Thank you. We have fixed the sentence.
Line309: can a more specific information be used to substitute the expression “strong level”
Thank you. We have fixed it.
Line315: “corroborating what is the knowledge”?
We have removed the confusing words in the sentence.
Line318: “ adenocarcinoma”
We have fixed it.
Line 322: delete “of”
We have fixed it.
Line336: which there is capital S (p53-Ab). Also define SCC-Ag
We have fixed it.
Table 1. See above. I believe the table need to be restructured considerably.
Ideally a meta-analysis of the various studies (or at least of the tumor types for which more data is available) could be attempted. Or as a minimum, an attempt to interpret the opposite results that are frequent in the table in light of sample size or method used or other variables.
We like the table. We have worked considerably on it. We think that a meta-analysis would be beyond the focus of the current review. However, we might consider it for a future paper. We took your suggestion to summarize the data in the conclusions to try to explain s-p53-Abs negative prognostic value is some type of solid tumors in the following terms: “That was the case for two studies of BC, two studies of NSCLC, one study of SCLC. On the contrary, more studies showed that the presence of s-p53-Abs predicted worse survival. Such result was mostly predominant in head and neck cancers -with five studies- followed -with CRC with 4 studies- and NSCLC with 3 studies.”
Line355: why is the dominant-negative effect mentioned at this point? Is there conclusive evidence that in primary cancer heterozygous for mutant p53, the mutant protein is expressed at very high levels?
We have removed the part related to dominant-negative as it could be confusing and there is no conclusive evidence about it. We only know that mutant p53 proteins often become stable and accumulate to very high levels in tumors.
Line 357-358: the sentence could be modified and rendered less speculative adding to the discussion on specific mutant p53 neoepitopes could be expanded. Consider also citing the Rosenber/Deniger JCI paper.
We have fixed the sentence. We have added the reference at the end of the paper as an argument of future therapies: “Additionally, it would be interesting investigating p53 antibodies recognizing more p53 hotspot mutations, including c.659A>G (p.Y220C) and c.733G>A (p.G245S), which have been uniquely discovered in human ovarian cancers [168]. Through targeted therapies it could be possible to target p53-mut in cancer cells, while sparing normal cells with the aim to improve efficacy and reduce toxicity to patients, respect to conventional therapies.”
Line374: please clarify in which sense mutant p53 act through mir-34. As it reads now the sentence does not seem to be correct.
We have changed it adding more information to better explain that phrase: “Of note, expressions of PD-L1 and p53 correlated in various types of cancers[163–167]. All these groups used immunohistochemistry, which detected p53-mut. Cortez et al showed that PD-L1 is a direct target of microRNA-34 (mir-34) using western blotting and luciferase assay. Using p53 knocked down mouse model they further showed that such regulation was dependent on the presence of p53-mut p53-mut downmodulates the expression of PD-L1 through mir-34 [163].
The conclusive statement can be refined: is the fact the only a small fraction of studies shows that p53 antibodies predict better survival the main message of Table 1 and the review and how does this connect with the expected therapeutic benefit of antibodies targeting mutant p53?
Interesting point. We have added the argument while fixing the final statement following your suggestion: “Although our review showed that in most of cases the presence of s-p53-Abs correlated with worse survival, it would be interesting investigating p53 antibodies recognizing p53 hotspot mutations, including c.659A>G (p.Y220C) and c.733G>A (p.G245S), which have been uniquely discovered in human ovarian cancers [168]. Through targeted therapies it could be possible to aim at treating p53-mut in cancer cells, while sparing normal cells with the aim to improve efficacy and reduce toxicity to patients, respect to conventional therapies.”
Reviewer 2 Report
The manuscript “Mutant p53 as an Antigen in Cancer Immunotherapy” reviews efforts to exploit tumor-derived mutant p53 antigens as a druggable target in cancer immunotherapy
After a concise introduction into p53 covering its discovery, its role in tumor suppression and the molecular mechanisms underlying the elevated (mutant) p53 levels in tumors, the authors describe the role of p53 as an antigen presented by cancer cells and the function of p53 in innate immune responses
In addition, the authors do a good job in providing a comprehensive overwiew of clinical studies investigating the prognostic value of serum anti-p53 antibodies in cancer, which is the major focus of the current manuscript.
Some minor points should be addressed before being acceptable for publication.
1.Personally, I have the impression that the last part of section2 (line 136 ff) and section3 which largely focus on the function(s) of (wt) p53 in innate immune responses are somewhat off-topic, considering that the focus of the manuscript should be on p53´s role as an antigen in adaptive immune responses. These could as well be briefly mentioned in the introduction, with references to other recent reviews such as the one the authors themselves provide (Blagih et al).
2. Line 114 : “p53 has been massively investigated over the last 40 years in the field of cancer biology; however, it could have happened in another way, investigating instead upon the p53 role in immunology and the immune response.”
This sounds too similar to a passage in a recent review by AJ Levine : “P53 and The Immune Response: 40 Years of Exploration—A Plan for the Future” and should be rephrased.
3. Line 116 : ”Nowadays, p53 is largely considered part of the innate immune system …..”
Could the authors provide a reference for this ?
“and, anytime soon, p53 tumor antigens might be a druggable target of immunotherapy”
These are somewhat disparate issues and should not necessarily be brought into context.
4. Line 109 : “We will briefly summarize p53-wt degradation using MDM2 as an example. Such mechanisms of degradation do not occur in p53-mut. Therefore this has been postulated as an explanation of why only p53-mut bearing patients result in the formation of p53-Abs (Figure 1).”
Line 127 : “As a consequence, missense mutations on the TP53 gene do not transactivate the MDM2 gene and the autoregulatory loop fails, resulting in higher levels of p53-mut protein in cancerous cells [35,55]”
Additional mechanisms (apart from additional E3 ligases) seem to underlie the tumor-specific upregulation of p53 levels, e.g. heat shock chaperon proteins, which should be mentioned.
5. Line 373 : “Of note, expressions of PD-L1 and p53 correlated in various types of cancers[155–159]. Additionally p53-mut downmodulates the expression of PD-L1 through microRNA-34.”
It should be discussed that in the mentioned papers showing a (positive) correlation between p53 and PD-L1 expression, what is being detected is probably mutant p53; otherwise the notion that PD-L1 can be negatively regulated via (wt)p53-miR-34a does not make sense.
6. Line 22: “the vast majority of published work on p53 antibodies in cancer patients use wild-type p53 as the antigen to detect these antibodies and it is unclear that they can recognize p53 mutants carried by cancer patients at all. We envision that an antibody targeting a specific mutant p53 will be effective therapeutically against the cancer carrying the exact mutant p53.”
Of note, a recent publication (Targeting mutant p53-expressing tumours with a T cell receptor-like antibody specific for a wild-type antigen, Low et al. 2019 ), showed that wtp53-specific TCRL antibodies can discriminate between wt and mutant p53 expressing cell lines based on p53 expression levels. (see also related publications)
While not directly related to the specificity of s-p53-Abs, these findings should at least be discussed.
Author Response
Peer-reviewer 2
The manuscript “Mutant p53 as an Antigen in Cancer Immunotherapy” reviews efforts to exploit tumor-derived mutant p53 antigens as a druggable target in cancer immunotherapy
After a concise introduction into p53 covering its discovery, its role in tumor suppression and the molecular mechanisms underlying the elevated (mutant) p53 levels in tumors, the authors describe the role of p53 as an antigen presented by cancer cells and the function of p53 in innate immune responses
In addition, the authors do a good job in providing a comprehensive overwiew of clinical studies investigating the prognostic value of serum anti-p53 antibodies in cancer, which is the major focus of the current manuscript.
Thank you.
Some minor points should be addressed before being acceptable for publication.
1.Personally, I have the impression that the last part of section2 (line 136 ff) and section3 which largely focus on the function(s) of (wt) p53 in innate immune responses are somewhat off-topic, considering that the focus of the manuscript should be on p53´s role as an antigen in adaptive immune responses. These could as well be briefly mentioned in the introduction, with references to other recent reviews such as the one the authors themselves provide (Blagih et al).
Thank you. We have removed those unnecessary parts as they were out of focus.
- Line 114 : “p53 has been massively investigated over the last 40 years in the field of cancer biology; however, it could have happened in another way, investigating instead upon the p53 role in immunology and the immune response.”
This sounds too similar to a passage in a recent review by AJ Levine : “P53 and The Immune Response: 40 Years of Exploration—A Plan for the Future” and should be rephrased.
Thank you. We rephrased it.
- Line 116 : ”Nowadays, p53 is largely considered part of the innate immune system …..”
Could the authors provide a reference for this ?
We have removed that phrase, as it was too strong.
“and, anytime soon, p53 tumor antigens might be a druggable target of immunotherapy”
These are somewhat disparate issues and should not necessarily be brought into context.
Thank you for the suggestion. We have removed that phrase with those two issues. We had addressed them in the discussions already.
- Line 109: “We will briefly summarize p53-wt degradation using MDM2 as an example. Such mechanisms of degradation do not occur in p53-mut. Therefore this has been postulated as an explanation of why only p53-mut bearing patients result in the formation of p53-Abs (Figure 1).”
Line 127: “As a consequence, missense mutations on the TP53 gene do not transactivate the MDM2 gene and the autoregulatory loop fails, resulting in higher levels of p53-mut protein in cancerous cells [35,55]”
Additional mechanisms (apart from additional E3 ligases) seem to underlie the tumor-specific upregulation of p53 levels, e.g. heat shock chaperon proteins, which should be mentioned.
Thank you for the correction. We have fixed the figure and added the following information of p53 accumulation: <<Accumulation of p53 can be triggered by many mechanisms, such as stress signals, DNA damage, nucleotide deprivation, DNA damage, mitogenic or oncogenic activation viral infection, heat shock proteins like HSP70/HSP40/HSP90, which in cancer cells form a multi-chaperone complex around p53-mut facilitating the unfolding of the p53-mut and its spontaneous folding to another conformation with different energy minimum [55–57] (Figure 1). In addition, the activity of p53 can be further enhanced by post-translational modifications working as positive or negative regulators [55,58–60].>>
- Line 373: “Of note, expressions of PD-L1 and p53 correlated in various types of cancers[155–159]. Additionally p53-mut downmodulates the expression of PD-L1 through microRNA-34.”
It should be discussed that in the mentioned papers showing a (positive) correlation between p53 and PD-L1 expression, what is being detected is probably mutant p53; otherwise the notion that PD-L1 can be negatively regulated via (wt)p53-miR-34a does not make sense.
Thank you. We have added << All these groups used immunohistochemistry, which detected p53-mut.>>
- Line 22: “the vast majority of published work on p53 antibodies in cancer patients use wild-type p53 as the antigen to detect these antibodies and it is unclear that they can recognize p53 mutants carried by cancer patients at all. We envision that an antibody targeting a specific mutant p53 will be effective therapeutically against the cancer carrying the exact mutant p53.”
Of note, a recent publication (Targeting mutant p53-expressing tumours with a T cell receptor-like antibody specific for a wild-type antigen, Low et al. 2019 ), showed that wtp53-specific TCRL antibodies can discriminate between wt and mutant p53 expressing cell lines based on p53 expression levels. (see also related publications)
While not directly related to the specificity of s-p53-Abs, these findings should at least be discussed.
Thanks. I have added what you suggested in the abstract together with the following sentence to corroborate what we were saying and give a glimpse to what will be our opinion in the discussions later on: “To corroborate such possibility, a recent study showed that a T cell-receptor-like (TCLR) antibody, initially made for a wild-type antigen, was capable of discriminating between p53-mut and p53-wt, specifically killing more cancer cells expressing p53-mut than p53-wt in vitro and inhibiting the tumour growth of mice injected with p53-mut cancer cells than mice with p53-wt cancer cells”.
Round 2
Reviewer 1 Report
The Authors have revised the manuscript, taking into account most of the suggestions raised.
In general, the manuscript flow has improved and most typos, have been corrected. Table 1 has not been modified, but the discussion has been expanded to compare and interpret some of the conflicting results that are present in the literature.
There are still a couple of issues that need attention
On page 3, section Transcriptional role of p53, relevant mutations and the mutant p53 GOF the Authors state that “approximately 80% of TP53 mutations …occur in six hotspots”
-I believe that this figure is wrong. The six hotspot mutations should not account for more than 30% of p53 mutations in cancer.
-I cannot see Figure 1 in the pdf. Figure 1 legend has been improved, but please check the sentence starting with: “current p53 antibodies”
Typos
Last line in page 5 “ consists”
Page 6, third line of last paragraph “nucleus”
Page 13 last sentence “specific p53 mutation recognized by the specific mutation??”
Author Response
The Authors have revised the manuscript, taking into account most of the suggestions raised.
In general, the manuscript flow has improved and most typos, have been corrected. Table 1 has not been modified, but the discussion has been expanded to compare and interpret some of the conflicting results that are present in the literature.
There are still a couple of issues that need attention
On page 3, section Transcriptional role of p53, relevant mutations and the mutant p53 GOF the Authors state that “approximately 80% of TP53 mutations …occur in six hotspots”
-I believe that this figure is wrong. The six hotspot mutations should not account for more than 30% of p53 mutations in cancer.
Thank you. We have fixed it to 30%.
-I cannot see Figure 1 in the pdf. Figure 1 legend has been improved, but please check the sentence starting with: “current p53 antibodies”
Thank you. We have corrected that phrase.
Typos
Last line in page 5 “ consists”
Thank you. We have corrected it.
Page 6, third line of last paragraph “nucleus”
Thank you. We have corrected it.
Page 13 last sentence “specific p53 mutation recognized by the specific mutation??”
Thank you. We have fixed it.